Changes in extremely hot days under stabilized 1.5°C and 2.0°C global warming
scenarios as simulated by the HAPPI multi-model ensemble
Michael Wehner[1*], Dáithí Stone[1], Dann Mitchell[2], Hideo Shiogama[3], Erich Fischer[4],
Lise S. Graff[5], Viatcheslav V. Kharin[6], Ludwig Lierhammer[7], Benjamin Sanderson[8],
Harinarayan Krishnan[1]
[1] Lawrence Berkeley National Laboratory, Berkeley, California 94720, USA
[2] University of Bristol, Bristol, United Kingdom
[3] National Institute for Environmental Studies, Tsukuba, Ibaraki 305-8506, Japan
[4] ETH Zurich, Switzerland
[5] Norwegian Meteorological Institute, Oslo, Norway
[6] Canadian Centre for Climate Modelling & Analysis, Victoria, British Columbia,
Canada
[7] German Climate Computing Center (DKRZ) Hamburg, Germany
[8] National Center for Atmospheric Research, Boulder, Colorado, USA
* Corresponding Author: mfwehner@lbl.gov
**Abstract**
The Half A degree additional warming, Prognosis and Projected Impacts (HAPPI)
experimental protocol provides a multi-model database to compare the effects of
stabilizing anthropogenic global warming of 1.5°C over preindustrial levels to 2.0°C
over these levels. The HAPPI experiment is based upon large ensembles of global
atmospheric models forced by sea surface temperature and sea ice concentrations
plausible for these stabilization levels. This paper examines changes in extremes of
high temperatures averaged over three consecutive days. Changes in this measure
of extreme temperature are also compared to changes in hot season temperatures.
We find that over land this measure of extreme high temperature increases from
about 0.5 to 1.5°C over present day values in the 1.5°C stabilization scenario
depending on location and model. We further find an additional 0.25 to 1.0°C
increase in extreme high temperatures over land in the 2.0°C stabilization scenario.
Results from the HAPPI models are consistent with similar results from the one
available fully coupled climate model. However, a complicating factor in interpreting
extreme temperature changes across the HAPPI models is their diversity of aerosol
forcing changes.

**Introduction**
The United Nations Framework Convention on Climate Change (UNFCCC)
challenged the scientific community to describe the impacts of stabilizing the global
mean temperature at its 21st Conference of Parties held in Paris in 2016. A specific
target of 1.5°C above preindustrial levels had not been seriously considered by the
climate modeling community prior to the Paris Agreement. Indeed, this level of
global warming is reached but then exceeded in most of the projections of the
Coupled Model Intercomparison Project (CMIP5), the source of much of our detailed
information about projected future climate change scenarios (Collins et al. 2013).
Analysis of these transient global climate model simulations as they pass through
1.5 and 2.0°C warmer temperatures than preindustrial estimates are not necessarily
descriptive of a stabilized climate due to the differential warming rates over land
and ocean regions of the planet. While pattern scaling (Tebaldi and Arblaster 2014)
of stabilized simulations at warmer levels may permit reasonable estimate of
surface air temperature and precipitation at the Paris Agreement targets, such
techniques have not been widely applied to other important output quantities from
climate models. Hence, custom simulations tailored to these 1.5 and 2.0°C targets
outside of the CMIP5 (and CMIP6) protocols are the most straightforward vehicles
for the scientific community to inform the UNFCCC.
Recently, the modeling group at the National Center for Atmospheric Research
(NCAR) performed simulations of the Community Earth System Model (CESM1)
suitably forced to stabilize to the Paris Agreement targets. Described in Sanderson
et al. (2017), these ocean-atmosphere coupled global simulations extend a previous
large ensemble (Kay et al. 2015) and provide a rather complete description of the
climate system at these stabilized levels and a path toward stabilization. However,
to more fully understand the model structural uncertainty in such projections,
efforts from additional modeling groups are necessary. In lieu of an internationally
coordinated extension to CMIP6 and to provide information prior to the publication
deadlines to the special report requested of the Intergovernmental Panel on Climate
Change, a limited number of modeling groups agreed to a simpler set of customized
simulations. The HAPPI experiment (Half A degree additional warming, Prognosis
and Projected Impacts) is based on the atmospheric components of CMIP5 models
forced by prescribed sea surface temperature (SST) and sea ice concentrations
(Mitchell et al. 2017). By replacing the ocean and sea ice components models with
prescribed values, simulation workflows are considerably simplified and
computational resource requirements reduced enabling the integration of larger
ensembles. SST and associated sea ice concentrations were specially constructed for
the HAPPI experimental protocol. Prescribed SSTs for the 1.5°C stabilization
scenario are obtained by adding the average climatological change over the periods
2006-2015 to 2091-2100 from the multi-model CMIP5 RCP2.6 ensemble to the
observed 2006-2015 SSTs. For the 2°C stabilization scenario, a weighted sum of the
RCP2.6 and RCP4.5 ensemble average changes over the same period is constructed
to be exactly 0.5°C warmer in the global mean than the 1.5°C experiment. Sea ice
concentrations are computed using an adapted version of the method described in
Massey (2017) by using observations of SST and ice to establish a linear relationship
between the two fields for the time period 1996-2015 and are consistent with the
HAPPI prescribed SST fields. While the changes to SST and sea ice concentrations
defining the stabilizations scenarios are identical for each HAPPI model, the actual
observations used come from a variety of well established sources chosen at the
discretion of the modeling groups. Details are further described in Mitchell et al.
88   (2017).
While HAPPI allows for large ensembles of multiple models to be compared, there
are tradeoffs to note in this simpler approach to modeling a stabilized climate
including the potential for radiative imbalance and inconsistencies between the
atmospheric state and the surface at the sea ice/ocean boundaries (Covey et al.
2004). Furthermore, while CMIP5 model differences in equilibrium climate
sensitivity are largely due to differences in ocean heat uptake (Collins et al. 2013),
important residual differences remain over land and global mean temperatures that
are not the same across the participating models. Finally, due to the prescribed SSTs
HAPPI does not account for different realizations of or potential changes in ocean
internal variability. The present study is confined to changes in extreme
temperatures over land simulated for the HAPPI project and defers these issues to
later analyses.
**Data and Methods**
Five modeling groups have submitted model output data to the HAPPI project that is
freely available to the public. Model #1 is the NCAR-DOE Community Atmosphere
Model version 4 (CAM4) coupled to the Community Land Model version 4 (CLM4)
with simulations contributed by ETH Zurich (Neale et al. 2011; Oleson et al. 2010).
Model #2 is the Canadian Fourth Generation Atmospheric Global Climate Model
(CanAM4) contributed by the Canadian Centre for Climate Modelling and Analysis
(von Salzen et al. 2013). Model #3 is ECHAM6.3 (Stevens et al. 2013), contributed by
the Max Planck Institute for Meteorology, Hamburg, Germany. It includes a modified
version of the land component (Reick et. al 2013). The soil hydrology is described by
a 5-layer scheme (Hagemann and Stacke 2015) instead of the bucket scheme used in
the CMIP5 version. Additionally, a high resolution (global 0.5° grid) hydrological
discharge model (Hagemann and Dümenil, 1997) is activated. Model #4 is the
MIROC5 model contributed by the National Institute for Environmental Studies,
Tsukuba, Japan and denoted as "MIROC5" (Shiogama et al. 2013, 2014). Model #5
(NorESM1) is an updated version of the Norwegian Earth System model version 1
(Bentsen et al. 2013, Iversen et al. 2013), contributed by the Norwegian Climate
Center. The NorESM1 is based on the NCAR Community Climate System Model
version 4 (Gent et al., 2011), but with a different ocean model and a modified
atmosphere component. The atmosphere model is based on the Community
Atmosphere Model version 4, but includes an advanced module for aerosols and
aerosol-cloud-radiation interactions (Kirkevåg et al. 2013). The version of the
NorESM1 used in the HAPPI project, NorESM1-Happi, additionally includes
improvements to wet snow albedo, and the atmospheric burden of soot (Iversen et
al, in prep.).
Aerosol forcings are not prescribed but left to the modeling groups to implement
based on their previous experience and simulations. The only constraint specified
by the HAPPI protocol is that the 1.5°C and 2°C use the same aerosol forcing.
Variations between model treatments in both the absolute magnitudes of the
aerosol forcing as well as their differences in the historical and stabilized scenarios
will prove to be an important factor in the changes in extreme temperatures.
An additional model result is also presented for comparison. The Community Earth
System Model (CESM1) is a fully coupled model that was not part of the HAPPI
project. However, 15 member ensembles of simulations were made under forcing
scenarios tailored to produce 1.5°C and 2°C stabilized climates (Sanderson et al.
2017). These simulations, while not directly comparable to the five HAPPI models,
provide additional context for extreme temperatures in stabilized low warming
scenarios.
The HAPPI experimental protocol was inspired by the "Climate of the 20th Century
Plus (C20C+) Detection and Attribution project" (Stone et al. 2017) and data from
both sets of simulations are available at the same website (portal.nersc.gov/c20c).
However, only output from the MIROC5 model was submitted to both projects. In
the HAPPI experimental protocol, the present day forcings and boundary conditions
are representative of the observed 2006-2015 state and is identical to that specified
in the C20C+ protocol over that period. HAPPI forcings for stabilized future
scenarios preserve the observed 2006-2015 interannual variability (Stone et al.
2017; Mitchell et al. 2017) but include appropriate changes derived from the CMIP5
RCP2.6 and RCP4.5 scenario simulations. Dates for these simulations are nominally
2106-2115 as atmospheric trace gas concentrations are scaled from the RCP's
protocol at 2095. Table 1 summarizes details of the model simulations used in this
study. Note that the ensemble sizes are exceptionally large for a publicly available
multi-model climate simulation dataset.
In this study, we examine the differences in changes in extreme temperatures from
the HAPPI simulations. In a companion paper, we examined such changes between
the actual and counterfactual (non-industrialized) simulations submitted to the
C20C+ project and this paper uses the same extreme value statistical methodologies
(Wehner et al. 2017). The annual maximum of the daily maximum temperature is
one of the 27 indices defined by the Expert Team on Climate Change Detection
Indices (ETCCDI) and is a robust indicator of extremely hot weather (Zhang et al.
2011). Called "*TXx*" by the ETCCDI and derived from "*tasmax*", the daily maximum
near surface air temperature in the CMIP5, this quantity is also known as "hot days"
because it is the hottest daytime temperature of the year. As in our previous work
on this topic (Tebaldi and Wehner 2016; Sanderson et al. 2017; Wehner et al. 2017),
we first calculate the running 3 day average of tasmax and compute its annual
maximum, denoted hereafter as *TX3x*, and then estimate its 20 year return values by
fitting stationary Generalized Extreme Value distributions. We have previously
found that while long period return values of *TX3x* are slightly smaller than for the
daily quantity, projected changes of the 3 day averages were considerably larger
(Tebaldi and Wehner 2016). For this study, where we are interested in the small
differences between the 1.5°C and 2.0°C stabilization levels, this point becomes
particularly important.
In this paper, we do not assess the HAPPI models' relative skill at reproducing
observed estimates of extremes temperatures. However, we note that this set of
models form the atmospheric components of several of the CMIP5 fully coupled
models. Sillman et al. (2014) did examine the CMIP5 model's performance in
simulating TXx and other ETCCDI measures. The coupled models corresponding to
these five HAPPI models spanned a large range of TXx errors when compared to
four different reanalyses. These model errors are presumably reduced when the
ocean is specified to its observed state.
As in our C20C+ analysis of anthropogenic extreme temperature changes, we
estimate 20-year return values by fitting the Generalized Extreme Value (GEV)
distribution by the methods of L-moments (Hosking and Wallis 1997). Assumptions
that the analyzed data is stationary and independent and identically distributed
(i.i.d) are necessary for this approach to be valid and are reasonable for the HAPPI
model output. A more detailed discussion of the rationale and limitations of these
assumptions for the C20C+ data is provided in Wehner et al. (2017) and the same
arguments hold for the HAPPI data. Originally introduced by Zwiers and Kharin
(1998) and Kharin and Zwiers (2000) to provide statistically rigorous projections of
future extreme temperature and precipitation, such GEV analyses, both stationary
and non-stationary, are now widespread throughout the literature including recent
assessment reports of the International Panel on Climate Change (Seneviratne et al.,
2012; Collins et al. 2013). The particulars of the details of the GEV analysis used in
this study are described in the Supplementary material of Tebaldi and Wehner
202    (2017).
By pooling the block maxima variable, TX3x, across both years and ensemble
members, the extreme value time series are equivalent in length to the product of
these two dimensions. As both the historical and stabilization periods are a decade,
this results in extreme value sample sizes for the HAPPI models that are 10 times
longer than the number of realizations in the 3[rd] column of table 1, ranging from 500
(MIROC5) to 5000 (CAM4). These large sample sizes of the HAPPI models (table 1)
ensure that uncertainty due to the fitting of statistical distribution is negligible. The
coupled model results (CESM1) are taken directly from Sanderson et al. (2017),
which used periods of three decades to compensate for the smaller ensemble size.

**Results**
We limit this study to reporting changes in 20 year return values of extreme
temperatures with the recognition that changes in longer period return values do
not differ greatly . This is principally due to the bounded nature of the fitted GEV
distributions and little difference in the width of these distributions  over most land
areas (Wehner et al. 2017). As changes in return periods for fixed thresholds are not
as stable to the choice of threshold values, any results we might report would be of
less general utility so we defer such to more targeted impact analyses. Figure 1
shows the changes over land in 20 year return values of the annual maximum of the
three day average of daily maximum surface air temperatures (*TX3x*) between the
1.5$^o$C stabilized scenario and the present day simulations. Of the HAPPI models,
MIROC5 exhibits the largest increases of the five HAPPI models exceeding 0.75$^o$C
nearly everywhere and even 1.25$^o$C over large regions. CAM4 and ECHAM6 exhibit
the smallest changes but do have hot spots in Asia. CanAM4, ECHAM6 and NorESM1
also show decreases or little increase over parts of the Amazon, but MIROC5 does
not. The fitted GEV parameters and hence these return value changes are extremely
robust to sample size uncertainty due to the large number of realizations in the
HAPPI database (Table 1). Standard errors determined by a bootstrap calculation
(Hoskins and Wallis 1997) are very small.  Results shown in figures 1-4 from the
coupled ocean-atmosphere model, CESM1, are shown for illustrative purposes only
and are not directly comparable to the HAPPI models as the experimental protocols
are necessarily different.

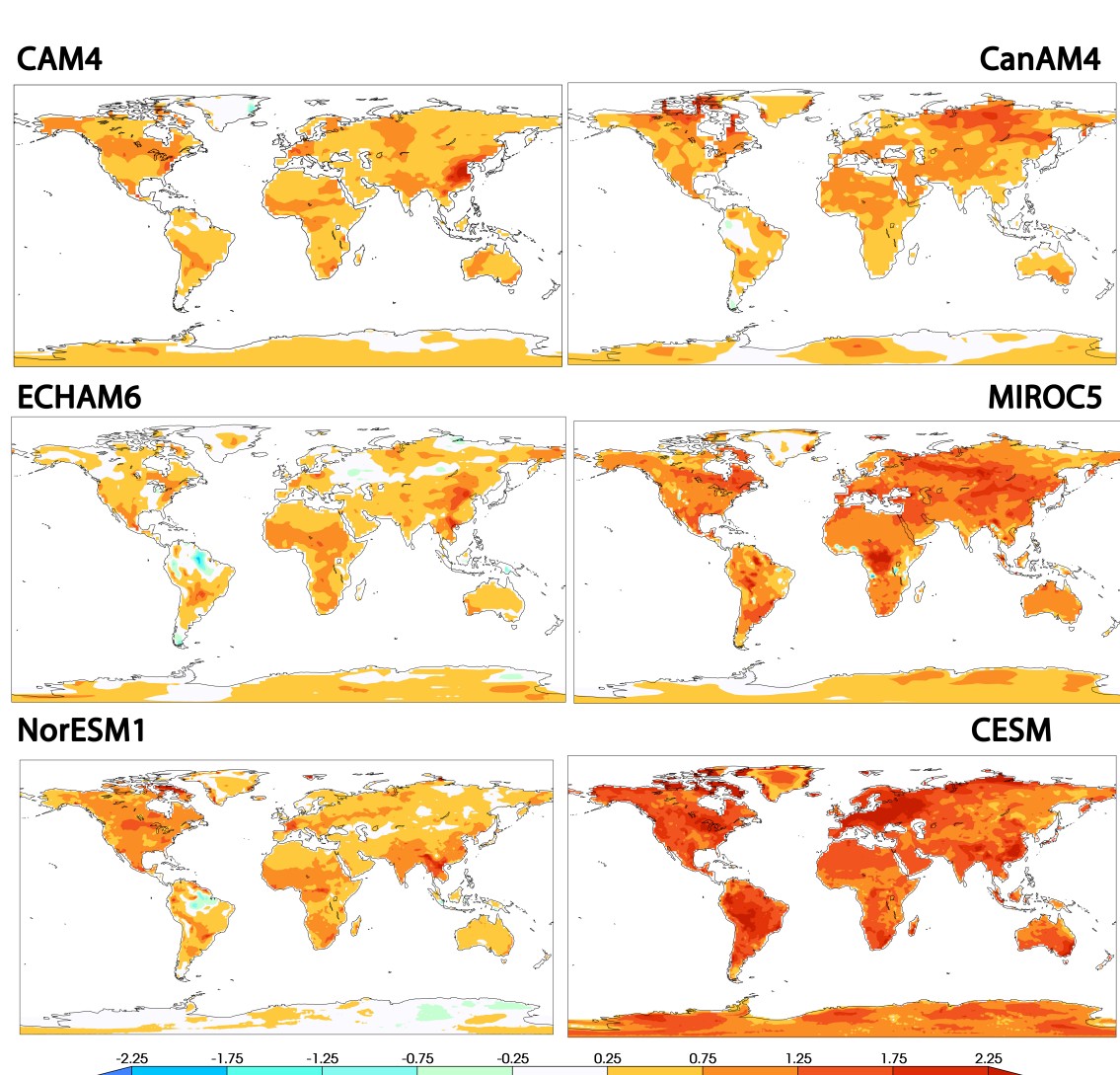

Figure 1. Change in 20 year return values ($^o$C) between the 1.5$^o$C and present day
HAPPI simulations of *TX3x*. Upper left: CAM4. Upper right: CanAM4. Middle left:
ECHAM6. Middle right: MIROC5. Lower left: NorESM1. Lower right: CESM.
The annual maximum of the daily high temperature is most likely to occur in the
summer over most of the world outside of the tropics. Figure 2 shows the difference
between the 1.5$^o$C stabilized scenario and the present day simulations of the
average surface air temperature in the hottest season, usually June-July-August in
the Northern Hemisphere and December-January-February in the Southern
Hemisphere. This much more spatially smooth average temperature change is quite
different from the extreme temperature change in other ways as well. Global land
average changes (shown in table 1) indicate that hot season temperatures generally
increase slightly more than extreme temperatures. However, there are significant
regional differences between models, as shown below in figure 6. Furthermore,
there are no regions of average temperature decreases. Average temperature
increases are always greater than 0.25°C.

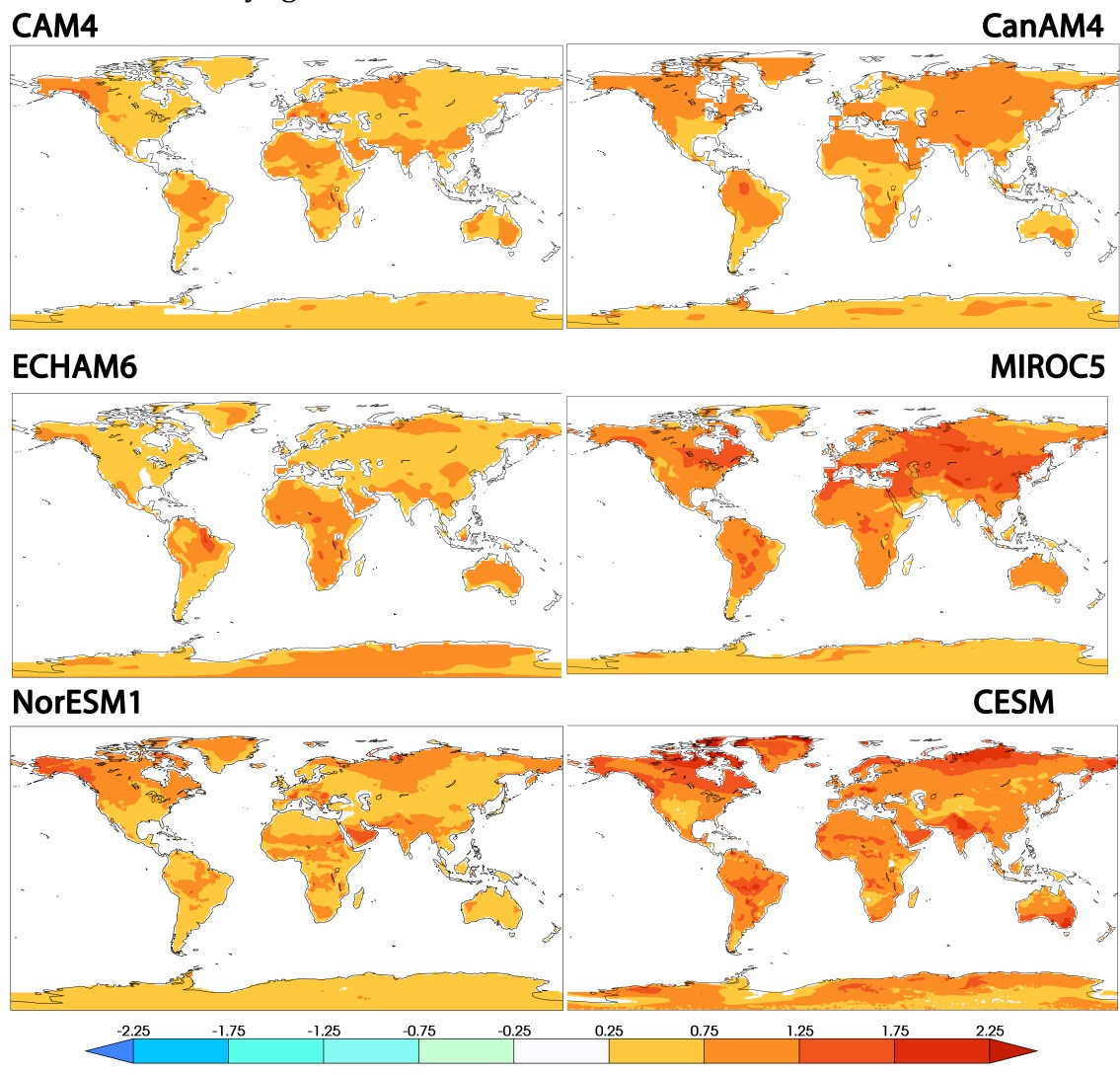

Figure 2. Differences in average hot season surface air temperature (°C) between the
1.5°C and present day HAPPI simulations. Upper left: CAM4. Upper right: CanAM4.
Middle left: ECHAM6. Middle right: MIROC5. Lower left: NorESM1. Lower right:
CESM.

Figure 3 shows the changes over land in 20 year return values of the annual
maximum of the three day average of daily maximum surface air temperatures
between the 2.0°C stabilized scenario and the present day simulations. As might be
expected, extreme temperature increases are larger than in the 1.5°C stabilized
scenario (figure 1). In this warmer scenario, most models produced no decreases in
extreme temperature. Only ECHAM6 has a small decrease in the Amazon. For
completeness, differences between the 2.0°C stabilized scenario and the present day
simulations of the average surface air temperature in the hottest season are shown
in the Appendix. As in the cooler scenario, global averaged land extreme
temperature differences are generally smaller than for the average hot season
temperature differences (Table 1). MIROC5, discussed in more detail below, is an
exception to this conclusion.

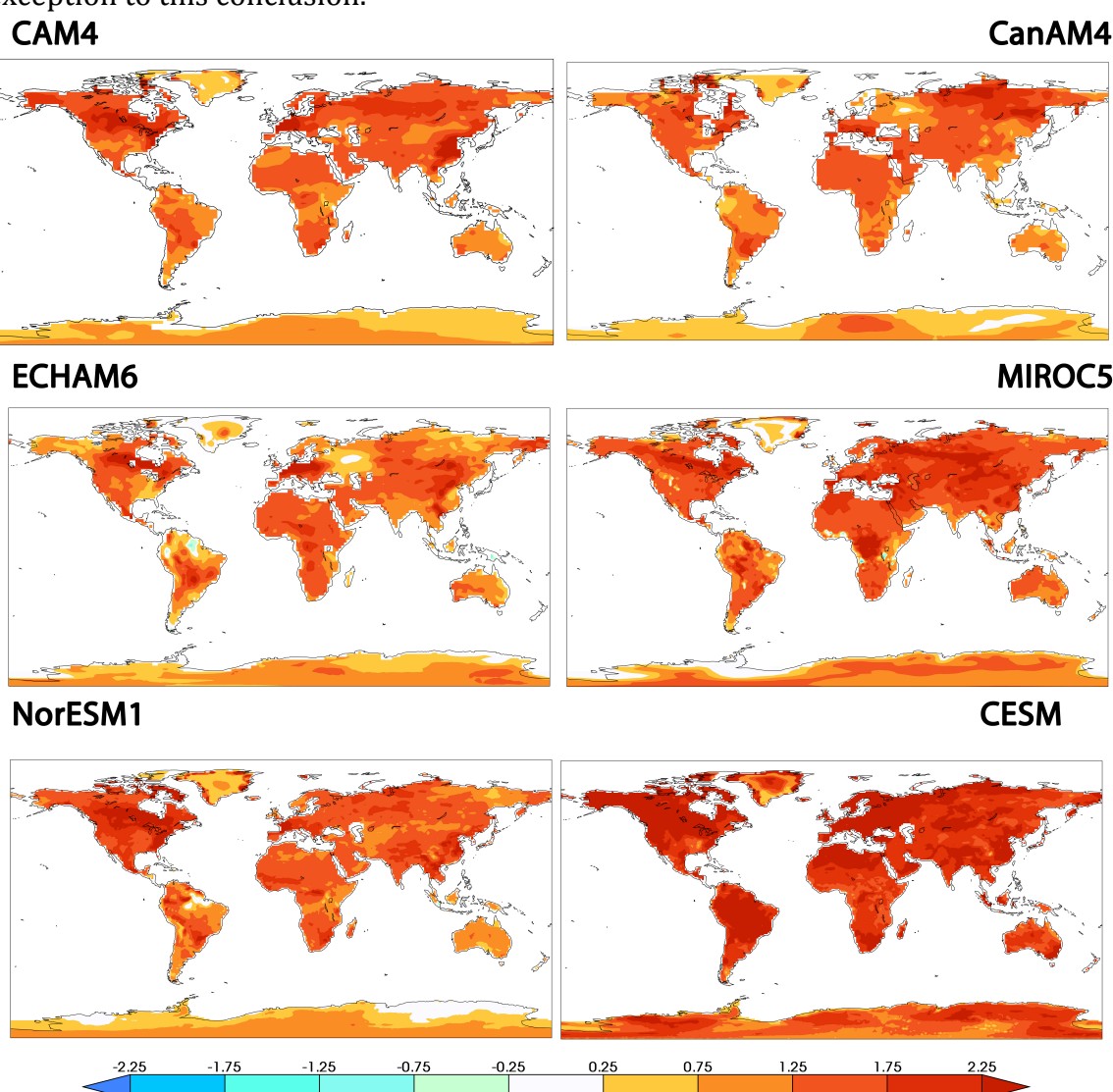


Figure 3. Change in 20 year return values (°C) between the 2.0°C and present day
HAPPI simulations of *TX3x*. Upper left: CAM4. Upper right: CanAM4. Middle left:
ECHAM6. Middle right: MIROC5. Lower left: NorESM1. Lower right: CESM.
Differences between the extreme temperatures of the 2.0°C and 1.5°C stabilized
scenarios are shown in figure 4. Global land average differences in extreme 3 day
hot temperatures range from about 0.5°C to 0.75°C (Table 1).

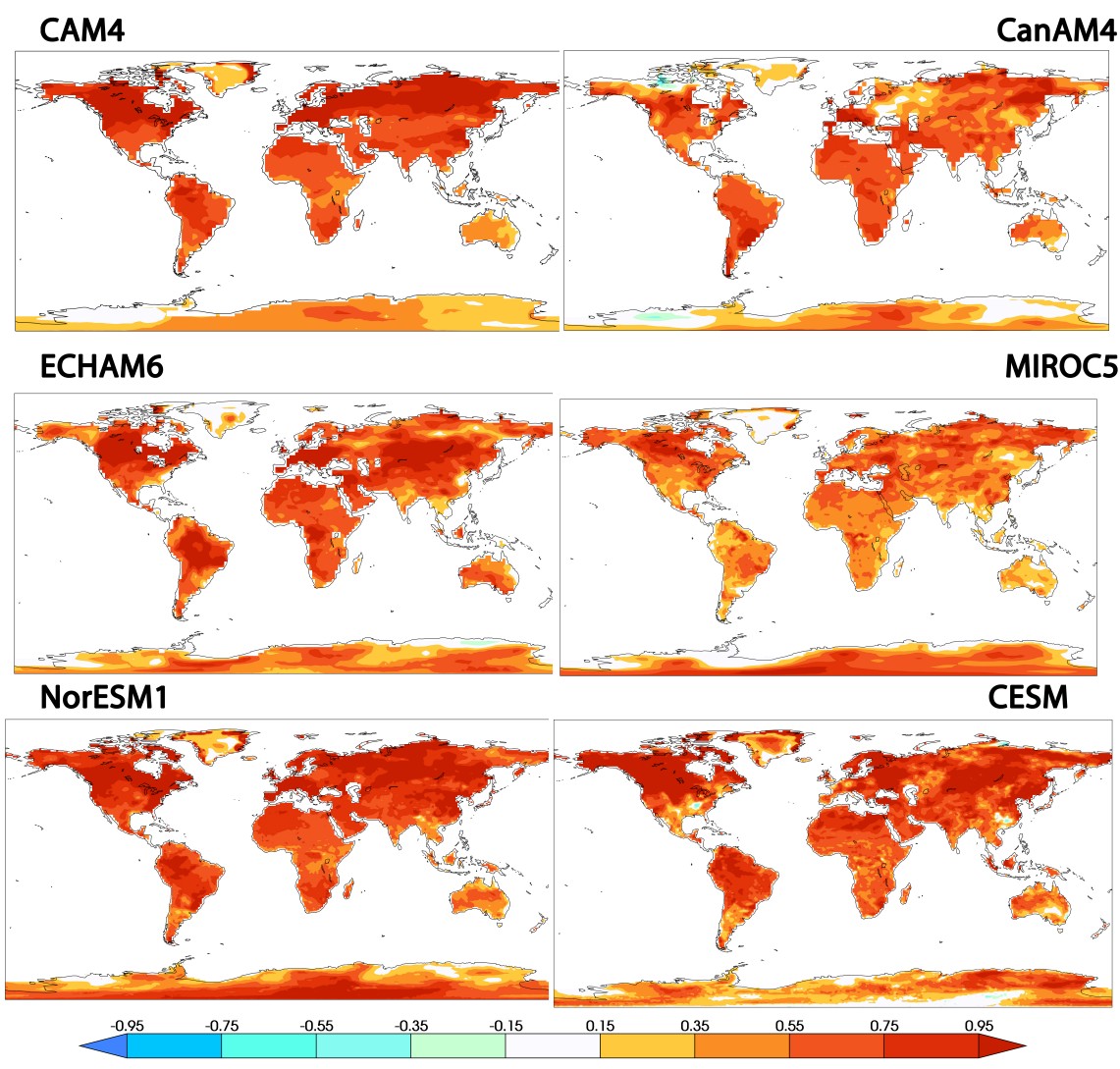

Figure 4: Differences in 20 year return values (°C) between the 2.0°C and 1.5°C
HAPPI simulations of *TX3x*. Upper left: CAM4. Upper right: CanAM4. Middle left:
ECHAM6. Middle right: MIROC5. Lower left: NorESM1. Lower right: CESM. Note that
the color scale covers a smaller range of temperature differences than for the
previous figures.
Standard errors obtained from the method of Hoskins and Wallis (1997) are shown to be
small in figure A3 of the appendix. Generally, these error estimates are less than $0.15^{\circ}$C
with the largest values towards the higher Northern latitudes. Variability in CanAM4 is
higher than the other HAPPI models but is generally less than $0.25^{\circ}$C. Standard error
estimates in the CESM are of a similar magnitude but are not directly equivalent. Most of
the changes in figures 1-4 are interpreted as at least at the *likely* level in the IPCC
calibrated language (Mastrandrea et al. 2010).
At this time, only a single coupled model, the CESM, has been run under $1.5^{\circ}$C and
$2^{\circ}$C stabilization conditions. Fortunately, a moderately sized ensemble of those
CESM simulations is available and analyzed in Sanderson et al. (2017) and shown in
the lower right panel of figures 1-4. The reference period from the "historical" run in
Sanderson et al. (2017) was earlier than for the HAPPI All-Hist and partly explains
the larger changes in the comparison between stabilization and current simulations
shown in figures 1-3. Although the method to simulate stabilized climates is quite
dissimilar between the HAPPI and the coupled model, differences between the $1.5^{\circ}$C
and $2.0^{\circ}$C stabilized CESM simulations of *TX3x* return values are quite similar to
CAM4, ECHAM6 and NorESM1 with global averages over land of $0.7^{\circ}$C or larger.
The MIROC5 is the only model for which results were submitted to both the C20C+
Detection and Attribution Project. In Wehner et al. (2017), we find that
anthropogenic aerosol forcing can play a critical role in heat wave attribution
statements. The MIROC5 experiments were run with a fully prognostic sulfate, black
carbon and organic carbon aerosol package forced by prescribed aerosol emissions.
In such experiments, aerosol concentrations can interact with the immediate
meteorology, leading in some regions to cooling, especially in events characterized
by persistent and stagnant air masses. This is indeed the case for the MIROC5 All-
Hist simulations compared to the C20C+ counterfactual simulations (Nat-Hist) of a
world without anthropogenic changes to the composition of the atmosphere. All-
Hist minus Nat-Hist extreme temperature from MIROC5 are replotted from Wehner
et al. (2017) in the top panel of figure 5 with a wider color scale to permit additional
comparison to the warmer stabilization scenarios. Decreases in extreme
temperatures are found in East Asia, the Congo and Eastern Europe that are
attributable to sulfate and organic carbon aerosol concentration differences for this
model. In the MIROC5 stabilization runs, sulfate and organic carbon aerosol
emissions are reduced according to the protocols of the RCP2.6 scenario. These
reductions allow the greenhouse gas contribution to temperature changes to
dominate leading to increases in these regions when comparing the stabilization
experiments to either the All-Hist and Nat-Hist MIROC5 experiments (figures 1,3
and 5). In fact, the cooling in these regions in the MIROC5 All-Hist experiment
results in localized hot spots when compared to the stabilization experiments
(figures 1 and 3). This is especially evident over the Congo in these figures.

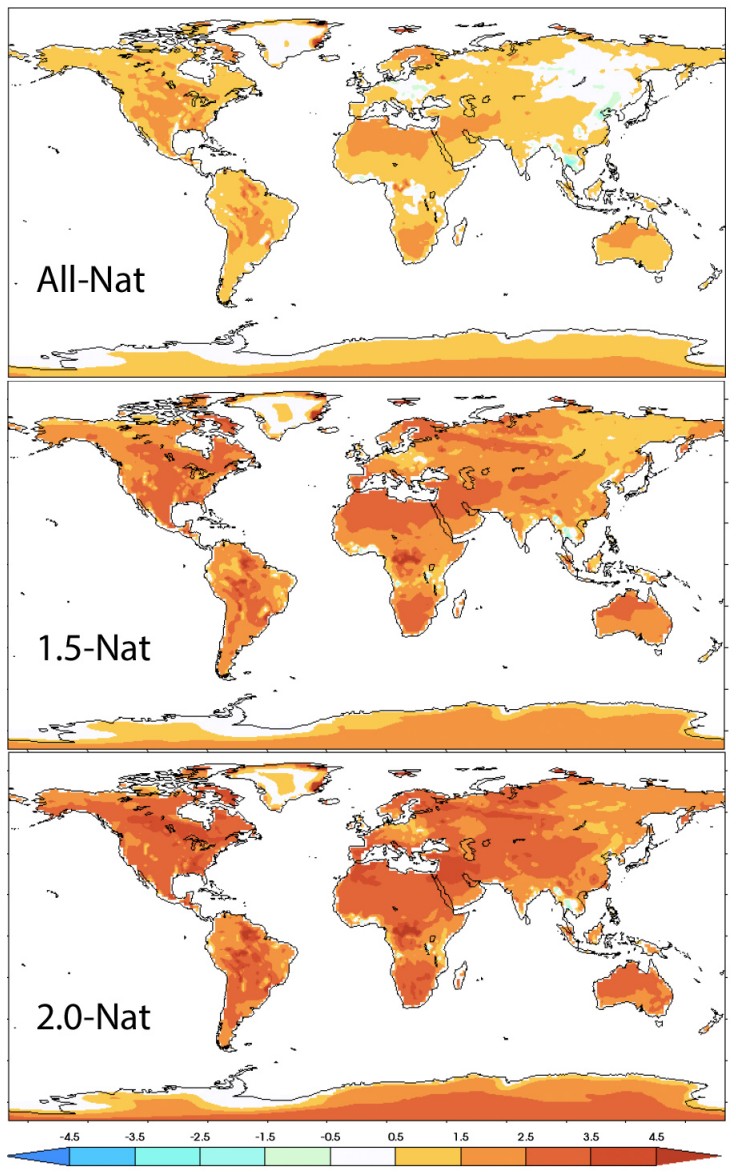

Figure 5: Change in 20 year return values ($^{o}$C) of *TX3x* between the C20C+ D&A
counterfactual simulation of a non-industrial world and the present day, 1.5$^{o}$C, 2.0$^{o}$C
HAPPI simulations for the MIROC5 model. Note that the color scale covers a larger
range of temperature differences than for the previous figures.
**Discussion**.
The Half A degree additional warming, Prognosis and Projected Impacts (HAPPI)
coordinated climate modeling experiments demonstrates that there are indeed
benefits in the form of reduced heat wave intensities associated with lower
stabilization targets. The large number of realizations permits estimation of these
reductions in heat wave magnitude to a high precision for each of the four
participating models. For two of the models (CanAM4, MIROC5), heat wave
differences between the 1.5$^{o}$C and 2$^{o}$C stabilization targets called for in the Paris
Agreement are close to 0.5$^{o}$C over large portions of the land mass.  The other 3
models showed reductions of approximately 0.75$^\circ$C over large regions of the land
mass.
The HAPPI experimental protocol was designed to explore roughly equal
increments of global warming with experiments of the present day, approximately
1$^\circ$C above preindustrial temperatures, compared to 1.5$^\circ$C and 2$^\circ$C above that
reference value. However, comparing the changes between the 1.5$^\circ$C stabilization
and present day to the changes between the 2.0$^\circ$C and 1.5$^\circ$C stabilizations reveals
profound differences across models in the pattern of warming, both in mean and
extreme temperatures. This is traceable in part to the unconstrained nature of the
aerosol forcings. Models vary in their response to aerosol forcing, especially in the
so-called "indirect" effect involving feedbacks with cloud nucleation processes.
However, more relevant to temperature extremes are that some models prescribed
atmospheric aerosol concentrations while others prescribed aerosols emissions. In
the former case, aerosol concentrations are slowly varying and independent of the
local meteorology. In the latter case, aerosol concentrations interact with the
meteorology and can be considerably larger than their climatological averages
during the stagnant conditions often associated with certain types of heat waves.
Higher aerosol concentrations lead to greater atmospheric reflectivity reducing
temperatures during such heat waves. In the RCP2.6, emissions of sulfate aerosols
are significantly reduced compared to the present day. Hence, the type of aerosol
treatment can affect magnitudes of the changes in simulated *TX3x* return values.
Relative to the non-industrial MIROC5 simulations, present day heat waves are
suppressed in eastern Asia and other areas where sulfate aerosol emissions are
currently high. As aerosol emissions in the stabilization scenarios are reduced from
present day levels, changes in heat waves are larger in these regions because of this
suppression. This is a possible explanation of some of the differences between
simulated *TX3x* return values in the stabilized scenario compared to the present
day. On the other hand, aerosol forcing in the two stabilizations scenarios are the
same leading to a more controlled comparison of the effects of increased
greenhouse gases. As a result, the differences between stabilization scenarios in
extreme temperature changes shown in figure 4 are less spatially heterogeneous
and more similar between models than changes relative to the present day (figures
1 and 3).
This relative uniformity in figure 4 suggests that pattern scaling of extreme
temperature changes in models in the CMIP5 (Coupled Model Intercomparison
Project) forced by the RCP2.6 forcings to the 1.5$^\circ$C stabilization target may be an
appropriate method to accurately estimate changes in extreme temperatures.
However, relating changes in average hot season temperatures to changes in long
period return values of *TX3x* is difficult in the low warming stabilization scenarios
considered here. Figure 6 shows the difference between changes in 20 year return
values of *TX3x* and changes in hot season average temperatures for the 2.0$^\circ$C
stabilization scenario relative to the historical period. There is no clear relationship
across models between changes in the middle of the temperature distribution to
changes in the tail. For instance, CanAM4 exhibits smaller changes in the *TX3x*
return values than in the hot season average. The couple model, CESM, exhibits the
opposite behavior. The other four HAPPI models are mixed with some regions
exhibiting greater changes in extreme temperatures but other regions exhibiting
lesser changes. The exaggerated effects on extreme temperatures of aerosol forcing
changes would tend to lead to larger changes in extreme temperature than for hot
season temperatures in the prescribed aerosol emission models since RCP2.6
reduces aerosol forcing. Hence, this mechanism may be partly responsible for the
heterogeneities in East Asia and the Congo of figure 6 but is not likely a factor for the
heterogeneities in North America and Europe.

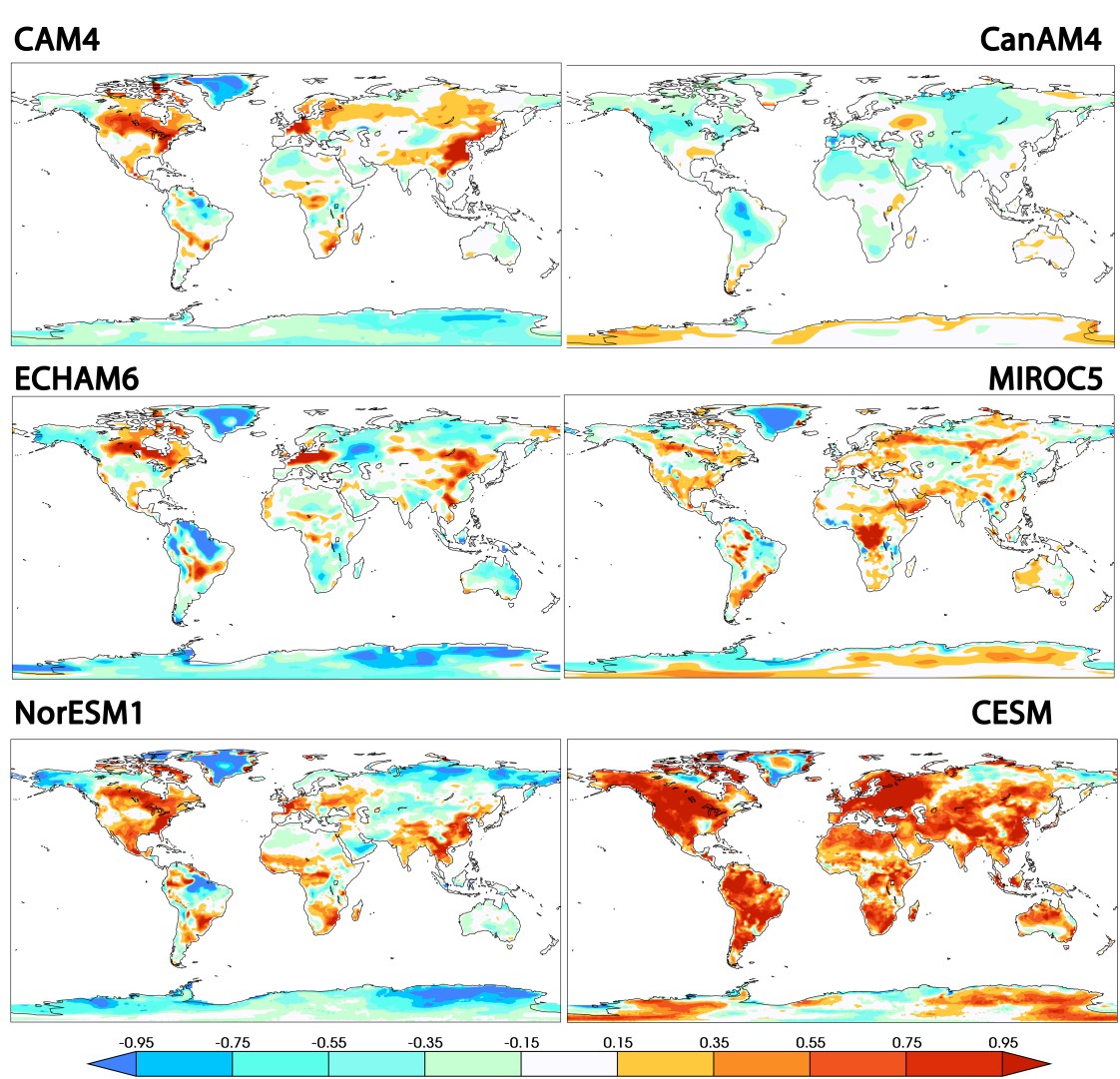

Figure 6: Differences between changes in 20 year return values of *TX3x* and changes
in hot season average temperatures ($^o$C) in the 2.0$^o$C HAPPI simulations. Upper left:
CAM4. Upper right: CanAM4. Middle left: ECHAM6. Middle right: MIROC5. Lower
left: NorESM1. Lower right: CESM

Land surface feedbacks offer another mechanism for different patterns of hot
season and extreme temperature changes. Evaporative cooling fueled by surface soil
moisture can locally reduce surface air temperatures (Seneviratne et al. 2010).
However, as the supply of surface soil moisture is limited, such temperature
reductions by evaporative cooling are also limited (Vogel et al 2017).  Hence during
extended periods without rain, dry conditions can enhance extreme high
temperatures. If this mechanism were important, one would expect changes in
extreme temperatures to be larger than average hot season temperature in regions
with moderate amounts of hot season rainfall.
Both the aerosol forcing and land surface feedback mechanisms would lead to
locally larger changes in extreme temperature compared to hot season
temperatures. We note that both mechanisms are diminished as greenhouse gas
forcing increases past those imposed by the HAPPI protocols. A physical mechanism
for the smaller extreme temperature changes in figure 6 is not readily apparent
although changes in large scale circulation are certainly a possibility (Koster et al
2014). Also, Fischer and Schär (2009) found a lengthening of the summer season in
parts of Europe that could also raise the average seasonal temperature more than
short duration extremes. In any event, we discount the possibility that these regions
of smaller extreme temperature changes are a result of statistical uncertainties due
to the large number of HAPPI realization in each ensemble.
The lack of a clear relationship in these models between hot season and extreme
temperature changes would seem to contradict that found by Seneviratne et al.
(2016) who found an approximately linear relationship between average regional
changes in TXx and changes in annual global mean temperature with slopes greater
than unity (i.e. extremes change more than the global mean). In general, we feel that
comparison of changes in very hot days to hot season average temperature changes
is more instructive than comparison to annual mean temperature changes in order
to more isolate relevant physical mechanisms of changes. For instance, changes in
albedo due to snowmelt may cause larger winter temperature changes than
temperature changes in other seasons.  However, the methods used to draw
conclusions from our study and Seneviratne et al. (2016) are too dissimilar to reveal
contradiction. Figure 6 shows a relationship between local temperatures for
individual models, while the results in Seneviratne et al. (2016) are a multi-model
re-expression of transient extreme temperature changes in terms of global mean
temperature instead of either time or greenhouse gas forcing.

**Conclusions**
Climate model experiments with identically prescribed sea surface temperature
(SST) and sea ice concentration such as presented here have a computational
advantage that permits large number of realizations enabling precise statistical
description of extreme temperatures. However, the limited number of models
participating in the HAPPI experiment does not sample the model structural
uncertainty as fully as the CMIP5 database of coupled models and the spread in
results presented here should not be interpreted as a complete representation of
the uncertainty in extreme temperature changes stabilized scenarios. Nonetheless,
although there is some amplification of extreme temperature differences relative to
average hot season temperature differences between the 1.5$^o$C and 2.0$^o$C
stabilization targets, this amplification does not appear to be dramatic.

**Acknowledgement**
This work was supported by the Regional and Global Climate Modeling Program of
the Office of Biological and Environmental Research in the Department of Energy
Office of Science under contract number DE-AC02-05CH11231. This document was
prepared as an account of work sponsored by the United States Government. While
this document is believed to contain correct information, neither the United States
Government nor any agency thereof, nor the Regents of the University of California,
nor any of their employees, makes any warranty, express or implied, or assumes any
legal responsibility for the accuracy, completeness, or usefulness of any information,
apparatus, product, or process disclosed, or represents that its use would not
infringe privately owned rights. Reference herein to any specific commercial
product, process, or service by its trade name, trademark, manufacturer, or
otherwise, does not necessarily constitute or imply its endorsement,
recommendation, or favoring by the United States Government or any agency
thereof, or the Regents of the University of California. The views and opinions of
authors expressed herein do not necessarily state or reflect those of the United
States Government or any agency thereof or the Regents of the University of
California.

Graff received support from the Norwegian Research Council, project no. 261821
(HappiEVA). HPC-resources for the NorESM model runs was provided in kind from
Bjerknes Centre for Climate Research and MET Norway. Storage for NorESM-data
was provided through Norstore/NIRD (ns9082k).

Shiogama was supported by the Integrated Research Program for Advancing
Climate Models (TOUGOU program) of the Ministry of Education, Culture, Sports,
Science and Technology (MEXT), Japan.

Lierhammer thanks Monika Esch, Karl-Hermann Wieners , Stefan Hagemann and
Thorsten Mauritsen from MPI-M for technical support with ECHAM6.3 and
Stephanie Legutke from DKRZ for guidance and advise. The project at DKRZ was
supported by funding from the Bundesministerium für Bildung und Forschung
(BMBF).

Bentsen, M., Bethke, I., Debernard, J. B., Iversen, T., Kirkevåg, A., Seland, Ø., Drange,
H., Roelandt, C., Seierstad, I. A., Hoose, C., Kristjánsson, J. E. Kristjánsson (2013) The
Norwegian Earth System Model, NorESM1-M - Part 1: Description and basic
evaluation of the physical climate. Geosci. Model Dev., 6 (3), 687-720,
doi:10.5194/gmd-6-687-2013.
Collins, M., R. Knutti, J. Arblaster, J.-L. Dufresne, T. Fichefet, P. Friedlingstein, X. Gao,
W.J. Gutowski, T. Johns, G. Krinner, M. Shongwe, C. Tebaldi, A.J. Weaver and M.
Wehner, 2013: Long-term Climate Change: Projections, Commitments and
Irreversibility. In: *Climate Change 2013: The Physical Science Basis. Contribution of*
*Working Group I to the Fifth Assessment Report of the Intergovernmental Panel on*
*Climate Change* [Stocker, T.F., D. Qin, G.-K. Plattner, M. Tignor, S.K. Allen, J. Boschung,
A. Nauels, Y. Xia, V. Bex and P.M. Midgley (eds.)]. Cambridge University Press,
Cambridge, United Kingdom and New York, NY, USA.
Covey, C., K. M. AchutaRao, P. J. Gleckler, T. J. Phillips , K. E. Taylor and M F. Wehner,
Coupled ocean-atmosphere climate simulations compared with simulations using
prescribed sea surface temperature:  Effect of a "perfect ocean". *Global and*
*Planetary Change* **41** (2004) 1-14
Fischer and Schär (2009) Future changes in daily summer temperature variability:
Driving processes and role for temperature extremes. Clim. Dynam. 33, 917–935.
doi:10.1007/s00382-008-0473-8
Gent, P. R., G. Danabasoglu, L. J. Donner, M. M. Holland, E. C. Hunke, S. R. Jayne, D. M.
Lawrence, R. B. Neale, P. J. Rasch, M. Vertenstein, P. H. Worley, Z.-L. Yang, and M.
Zhang, 2011: The Community Climate System Model Version 4. J. Clim., 24 (19),
4973-4991, doi:10.1175/2011JCLI4083.1.
Hageman, S. and T. Stacke (2015) Impact of the soil hydrology scheme on simulated
soil moisture memory. Clim. Dyn., 44, 1731-1750.
Hageman, S. and  L. Dümenil (1997) A parameterization of the lateral waterflow for
the global scale. Clim. Dyn., 14, 17-31.
Hosking, J. R. M., J. R. Wallis (1997) Regional Frequency Analysis: An Approach
Based on L-Moments. Cambridge University Press, 244 pages
IPCC, 2001: Climate Change 2001: The Scientific Basis. Contribution of Working
Group I to the Third Assessment Report of the Intergovernmental Panel on Climate
Change. Appendix 12.3. [Houghton, J.T., Y. Ding, D.J. Griggs, M. Noguer, P.J. van der
Linden, X. Dai, K. Maskell, and C.A. Johnson (eds.)]. Cambridge University Press,
Cambridge, United Kingdom and New York, NY, USA, 881pp

Iversen, T., Bentsen, M., Bethke, I., Debernard, J. B., Kirkevåg, A., Seland, Ø., Drange,
H., Kristjansson, J. E., Medhaug, I., Sand, M., Seierstad, and I. A., 2013: The Norwegian
Earth System Model, NorESM1-M - Part 2: Climate response and scenario
projections. Geosci. Model Dev., 6 (2), 389-415, doi:10.5194/gmd-6-389-2013.
Kay, J., Deser, C., Phillips, A., Mai, A., Hannay, C., Strand, G., Arblaster, J., Bates, S.,
Danabasoglu, G., Edwards, J., Holland, M., Kushner, P., Lamarque, J.-F., Lawrence, D.,
Lindsay, K., Middleton, A., Munoz, E., Neale, R., Oleson, K., Polvani, L., Vertenstein, M.:
The Community Earth System Model (CESM) large ensemble project: a community
resource for studying climate change in the presence of internal climate variability,
B. Am. Meteorol. Soc., 96, 1333–1349, 2015
Kharin, V.V., F.W. Zwiers (2000) Changes in the extremes in an ensemble of
transient climate simulation with a coupled atmosphere–ocean GCM. J. Clim. 13 ,
3760–3788 (2000
Kirkevåg, A., T. Iversen, Ø. Seland, C. Hoose, J. E. Kristjánsson, H. Struthers, A. M. L.
Ekman, S. Ghan, J. Griesfeller, E. D. Nilsson, and M. Schulz, 2013: Aerosol-climate
interactions in the Norwegian Earth System Model – NorESM1-M. Geosci. Model
Dev., 6 (1), 207–244, doi:10.5194/gmd-6-207-2013.
Koster, R. D., Y. Chang, S. D. Schubert (2014), A mechanism for land-atmosphere
feedback involving planetary wave structures, J. Clim., 27, 9290–9301.
Mastrandrea, M.D., C.B. Field, T.F. Stocker, O. Edenhofer, K.L. Ebi, D.J. Frame, H. Held,
E. Kriegler, K.J. Mach, P.R. Matschoss, G.-K. Plattner, G.W. Yohe, and F.W. Zwiers,
2010:  Guidance Note for Lead Authors of the IPCC Fifth Assessment Report on
Consistent Treatment of Uncertainties . Intergovernmental Panel on Climate Change
(IPCC). Available at <http://www.ipcc.ch>
Mitchell, D., AchutaRao, K., Allen, M., Bethke, I., Forster, P., Fuglestvedt, J., Gillett, N.,
Haustein, K., Iverson, T., Massey, N., Schleussner, C.-F., Scinocca, J., Seland, Ø.,
Shiogama, H., Shuckburgh, E., Sparrow, S., Stone, D., Wallom, D., Wehner, M., and
Zaaboul, R.: Half a degree Additional warming, Projections, Prognosis and Impacts
(HAPPI): Background and Experimental Design (2017). Geosci. Model Dev. 10, 571-
583, https://doi.org/10.5194/gmd-10-571-2017, 2017
Neale, R. B., and Coauthors, 2011: Description of the NCAR Community Atmosphere
Model (CAM4). NCAR Tech. Note NCAR/TN-485+STR, National Center for
Atmospheric Research, Boulder, CO 120 pp.
Oleson, K.W., D.M. Lawrence, G.B. Bonan, M.G. Flanner, E. Kluzek, P.J. Lawrence, S.
Levis, S.C. Swenson, P.E. Thornton, A. Dai, M. Decker, R. Dickinson, J. Feddema, C.L.
Heald, F. Hoffman, J.-F. Lamarque, N. Mahowald, G.-Y. Niu, T. Qian, J. Randerson, S.
Running, K. Sakaguchi, A. Slater, R. Stockli, A. Wang, Z.-L. Yang, Xi. Zeng, and Xu.
Zeng, 2010: Technical Description of version 4.0 of the Community Land Model
(CLM). NCAR Tech. Note NCAR/TN-478+STR, National Center for Atmospheric
Research, Boulder, CO, 257 pp.
Reick, C. H., et al. (2013) Representation of natural and anthropogenic land cover
change in MPI-ESM, J. Adv. Modeling Earth Sys. 5, 459-482.
Tebaldi C., M. Wehner (2016) Benefits of mitigation for future heat extremes under
RCP4.5 compared to RCP8.5. Climatic Change. DOI:10.1007/s10584-016-1605-5
Sanderson, B.M., Y. Xu, C. Tebaldi, M. Wehner, B. O'Neill, A. Jahn, A. G. Pendergrass, F.
Lehner, W. G. Strand, L. Lin, R. Knutti, and J. F. Lamarque (2017) Community Climate
Simulations to assess avoided impacts in 1.5 °C and 2 °C futures. Earth Sys. Dyn. 8,
827-847. https://doi.org/10.5194/esd-8-827-2017
Seneviratne, S.I., T. Corti, E. L. Davin, M. Hirschi, E. B. Jaeger, I. Lehner, B. Orlowsky,
A. J. Teuling (2010) Investigating soil moisture–climate interactions in a changing
climate: A review, Earth-Science Reviews, 99, Pages 125-161. ISSN 0012-8252,
https://doi.org/10.1016/j.earscirev.2010.02.004.
Seneviratne, S. I., et al., 2012: Changes in climate extremes and their impacts on the
natural physical environment. In: Managing the Risks of Extreme Events and
Disasters to Advance Climate Change Adaptation. A Special Report of Working
Groups I and II of the Intergovernmental Panel on Climate Change (IPCC) [C. B. Field,
et al. (eds.)]. Cambridge University Press, Cambridge, United Kingdom, and New
York, NY, USA, pp. 109–230.
Shiogama, H., M. Watanabe, Y. Imada, Y., M. Mori, M. Ishii, M. Kimoto (2013) An
event attribution of the 2010 drought in the South Amazon region using the MIROC5
model. Atmos. Sci. Lett. 14, 170-175
Shiogama, H., M. Watanabe, Y. Imada, Y., M. Mori, Y. Kamae, M. Ishii, M. Kimoto
(2014) Attribution of the June-July 2013 heat wave in the southwestern United
States, SOLA, 10, 122-126, doi10.2151/sola.2014-025
Sillmann, J., V. V. Kharin, X. Zhang, F. W. Zwiers, and D. Bronaugh (2013), Climate
extremes indices in the CMIP5 multimodel ensemble: Part 1. Model evaluation in the
present climate, J. Geophys. Res. Atmos., 118, 1716-1733, doi:10.1002/jgrd.50203.
Stark, J. D., C. J. Donlon, M. J. Martin, M. E. McCulloch, (2007) OSTIA: An operational,
high resolution, real time, global sea surface temperature analysis system, in:
Oceans 2007-Europe, 1–4, 2007
Stevens, B., et al. (2013) Atmospheric component of the MPI-M Earth System Model:
ECHAM6, J. Adv. Modeling Earth Sys., , 5, 146-172.
Tebaldi, C. & Arblaster, J.M. Pattern scaling: Its strengths and limitations, and an
update on the latest model simulations Climatic Change (2014) 122: 459.
https://doi.org/10.1007/s10584-013-1032-9
Vogel, M. M., R. Orth, F. Cheruy, S. Hagemann, R. Lorenz, B. J. J. M. van den Hurk, S. I.
Seneviratne (2017), Regional amplification of projected changes in extreme
temperatures strongly controlled by soil moisture-temperature feedbacks, Geophys.
Res. Lett., 44, 1511–1519, doi:*10.1002/2016GL071235.*
von Salzen, K., J. F. Scinocca, N. A. McFarlane, J. Li, J. N. S. Cole, D. Plummer, D.
Verseghy, M. C. Reader, X. Ma, M. Lazare, L. Solheim (2013) The Canadian Fourth
Generation Atmospheric Global Climate Model (CanAM4). Part I: Representation of
Physical Processes. Atmosphere-Ocean Vol. 51, 104–125
Wehner, M. F., D. Stone, H. Shiogama, P. Wolski, A. Ciavarella, N. Christidis, H.
Krishnan (2017) Early 21st century anthropogenic changes in extremely hot days as
simulated by the C20C+ Detection and Attribution multi-model ensemble. Submitted
to *Weather and Climate Extremes* special C20C+ issue.
Zhang, X., L. Alexander, G.C. Hegerl, P. Jones, A.K. Tank, T.C. Peterson, B. Trewin, F.W.
Zwiers (2011), Indices for monitoring changes in extremes based on daily
temperature and precipitation data. WIREs Clim Change, 2: 851–870.
doi:10.1002/wcc.147

Zwiers, F.W., V.V. Kharin,(1998)Changes in the extremes of the climate simulated by
CCC GCM2 under CO2 doubling. J. Clim. 11 , 2200–2222

Appendix

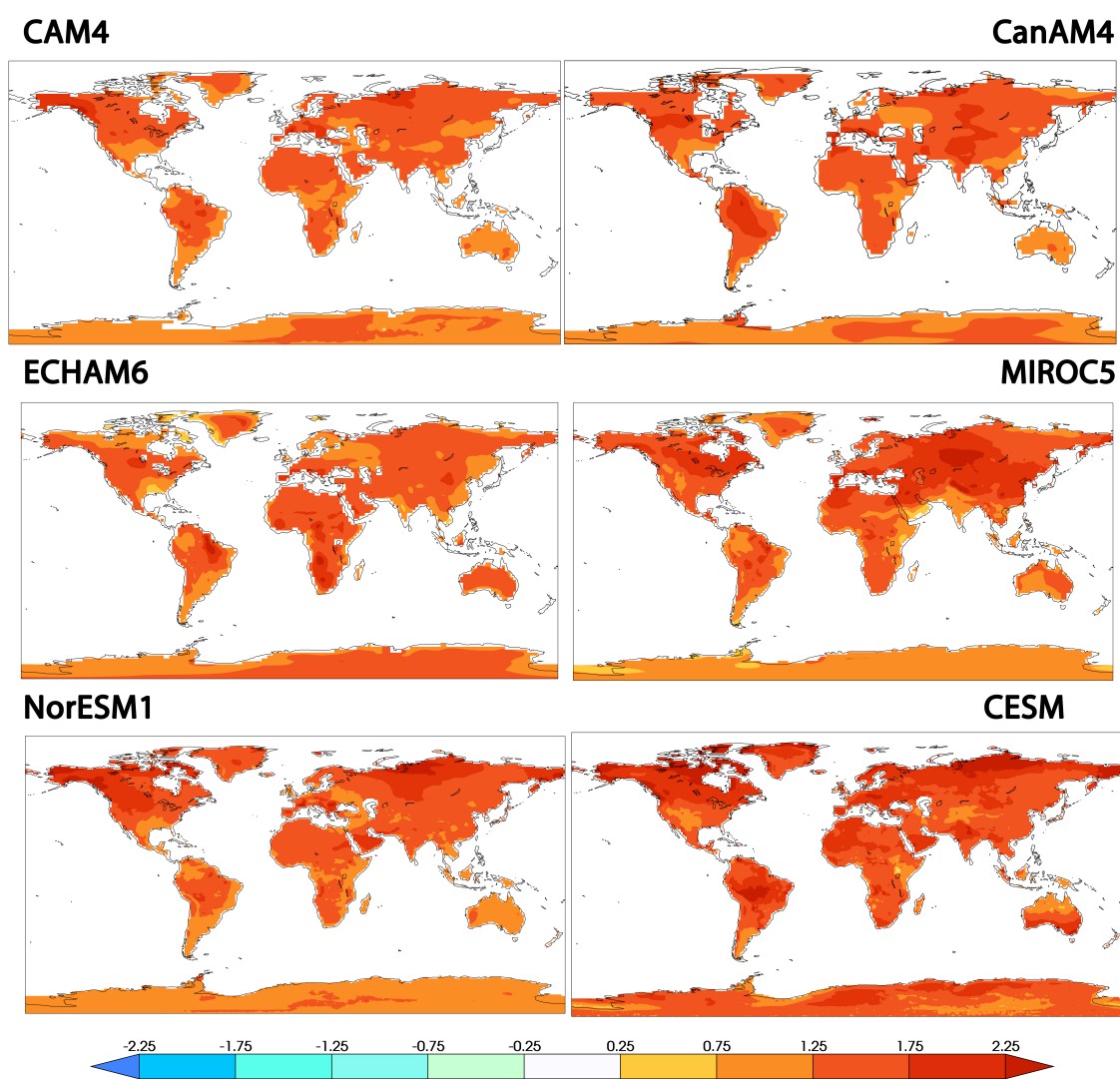


Figure A1. Differences in average hot season surface air temperature ($^o$C) between
the 2.0$^o$C and present day HAPPI simulations. Upper left: CAM4. Upper right:
CanAM4. Middle left: ECHAM6. Middle right: MIROC5. Lower left: NorESM1. Lower
Right. CESM.


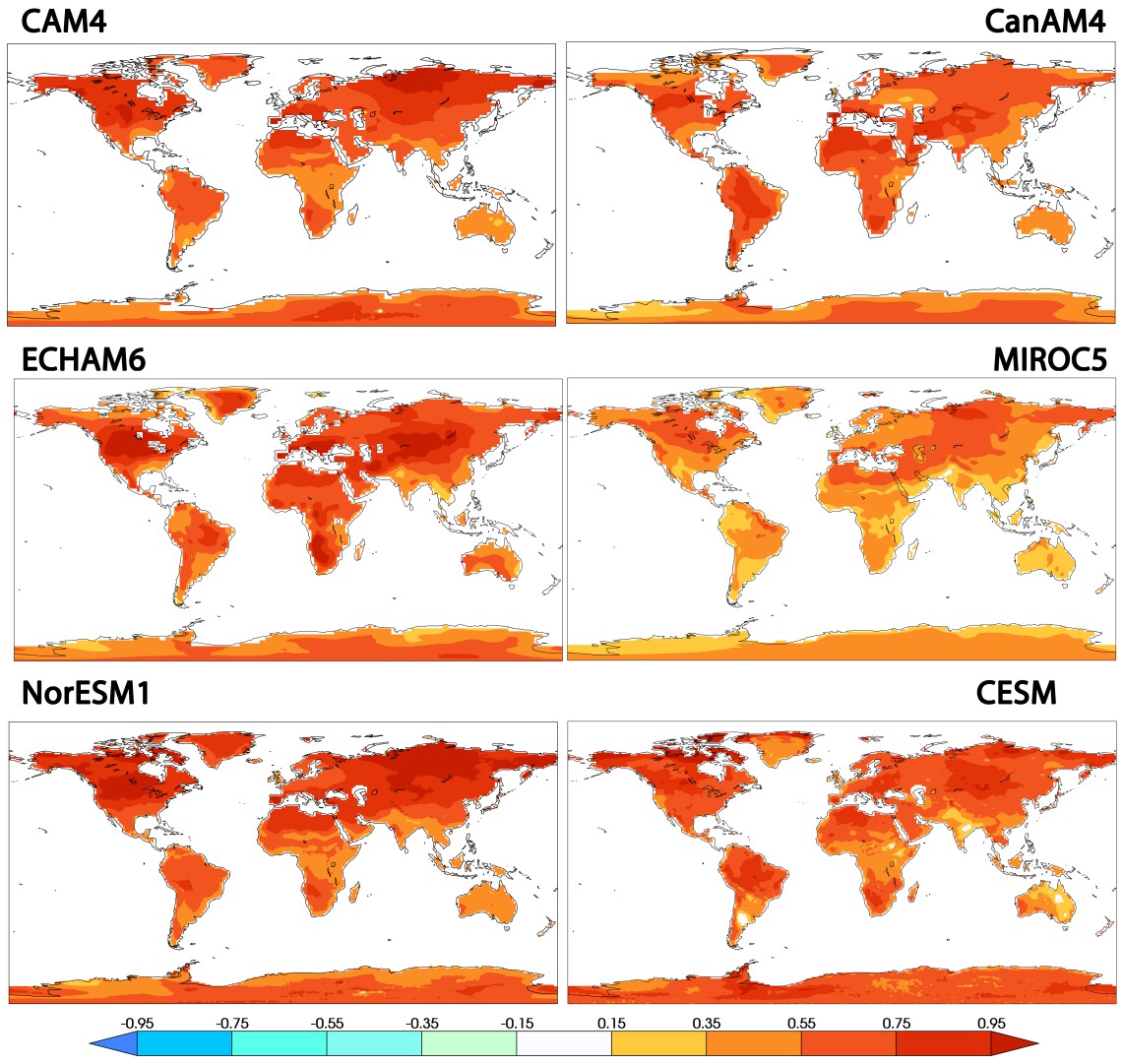

Figure A2: Differences in average hot season surface air temperature (°C) between
the 2.0°C and 1.5°C HAPPI simulations. Upper left: CAM4. Upper right: CanAM4.
Middle left: ECHAM6. Middle right: MIROC5. Lower left: NorESM1. Lower Right.
CESM.

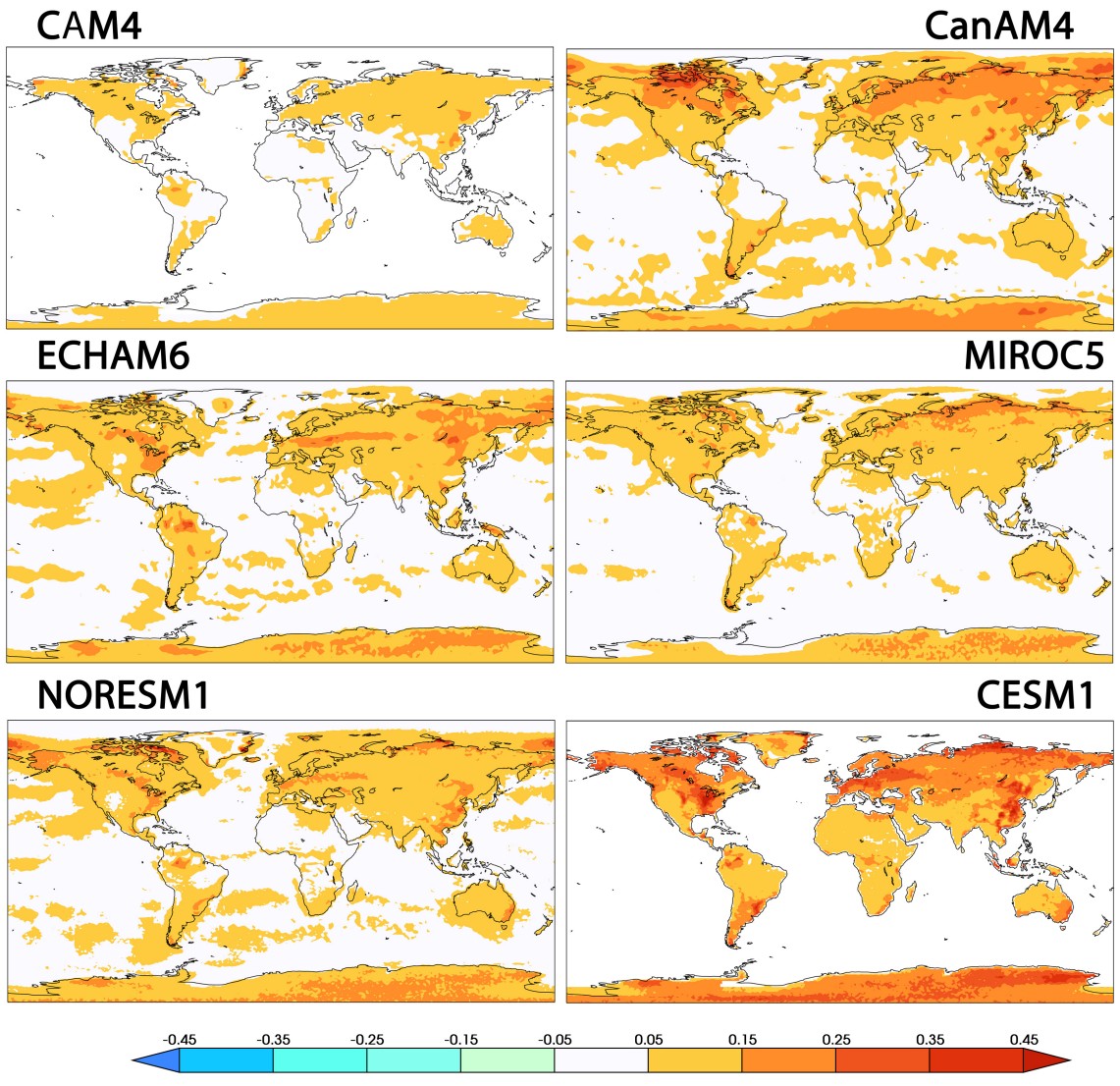

Figure A3: Standard error estimates of 20 year return values of *TX3x* ($^o$C) in the
1.5$^o$C or 2.0$^o$C HAPPI simulations. Upper left: CAM4. Upper right: CanAM4. Middle
left: ECHAM6. Middle right: MIROC5. Lower left: NorESM1. Lower Right. CESM.

| Model | Resolution (#lat X #long) | Number of realizations (Nat-Hist/All-Hist /Plus15/Plus20) | Global land average change in hot season mean temperature ($^o$C) | | | | Global land average change in very extreme temperature ($^o$C) | | | |
|---|---|---|---|---|---|---|---|---|---|---|
| | | | All-Hist Minus Nat-Hist | Plus15 minus All-Hist | Plus20 minus All-Hist | Plus20 minus Plus15 | All-Hist Minus Nat-Hist | Plus15 minus All-Hist | Plus20 minus All-Hist | Plus20 minus Plus15 |
| CAM4 | 96x144 | --/500/500/500 | -- | 0.69 | 1.33 | 0.64 | -- | 0.64 | 1.34 | 0.71 |
| CanAM4 | 64x128 | --/100/100/100 | -- | 0.80 | 1.40 | 0.60 | -- | 0.66 | 1.23 | 0.58 |
| ECHAM6-3-LR | 96x192 | --/100/100/100 | -- | 0.70 | 1.36 | 0.65 | -- | 0.48 | 1.12 | 0.69 |
| MIROC5 | 128x256 | 50/50/50/50 | 1.03 | 1.02 | 1.46 | 0.44 | 0.99 | 1.01 | 1.49 | 0.48 |
| NorESM1 | 192x288 | --/125/125/125 | -- | 0.72 | 1.41 | 0.70 | -- | 0.61 | 1.37 | 0.77 |
| CESM1 | | --/40/10/10 | -- | 0.89 | 1.39 | 0.50 | -- | 1.45 | 2.2 | 0.74 |

Table 1. Details of the HAPPI models used in this study. The number of realizations is for each part of the numerical experiment separately as used in this study. For some individual years of the All-Hist and Nat-Hist simulations, additional realizations may be available. The two rightmost columns shows the globally averaged difference between selected combinations of the hot season temperature and the 20 year return value of the annual maximum 3 day average daily

maximum surface air temperature (*TX3x*) over land.  "Hot season" is defined as the maximum of JJA and DJF averages. Plus2.0 denotes the 2$^o$C stabilization scenario, Plus1.5 denotes the 1.5$^o$C stabilization scenario. Note that CESM1 is not part of the HAPPI experiment but a fully coupled ocean-atmosphere climate model that has been run with emissions scenarios consistent with both targets. The CESM1 experiments are roughly comparable to the HAPPI experiment but not exactly the same forcing or reference period.