# Peer review of "Earth System Dynamics Discussions"

_Earth System Dynamics, 2017_

## Referee Comment (RC1) · Anonymous Referee #1 · 5 Dec 2017

GENERAL COMMENTS:

In this manuscript, Wehner et al. use a novel set of climate model ensembles to compare extreme temperatures under scenarios with 1.5 and 2 degrees of stabilized warming. I think this manuscript presents a useful and relevant assessment of the differences in a metric of extreme temperatures between the two scenarios, but would benefit from additional explanations, particularly of the methods.

SPECIFIC COMMENTS:

- In the abstract, I think it would be worth mentioning the range of increases compared to the All-Hist scenarios as well.

- Lines 72-75: Is the prescribed SST field determined from the CMIP5 ensemble mean?

- Lines 144-146: I'm having trouble seeing the importance of this sentence.

- Can the authors expand the explanation of the benefit of using 3-day averages?

- In line 163, it is mentioned that the 20 year return values of TX3x are estimated "using a block maxima technique", but as the TX3x values are block maxima, this description provides little information. It would probably be better to say "using the Generalized Extreme Value distribution" instead.

- It was not clear how the ensembles contributed to the values calculated and plotted in this paper. Were the block maxima pooled across the realizations before fitting the GEV distribution or are the ensemble mean return periods plotted? Additionally, it would be helpful if the number of values used to fit the GEV distributions were stated.

- Line 190: The first paragraph of the Results section can be removed, as much of this seems confusing and distracting and the relevant information is presented again at the beginning of the next paragraph.

- In all of the six-panel figures, the authors should make a distinction between CESM and the HAPPI models. I accept an argument for including the CESM model information, but these simulations are not directly comparable to the others.

- Line 233: Perhaps mention the later discussion of the changes in mean vs extremes.

- For Figure 6, the caption description does not match what is listed in the text (at line 362), which causes confusion in the understanding of what is being plotted.

- Line 419-420: Has this been tested? If not, perhaps rephrase to present this statement with less confidence.

- The "Conclusions" section is lengthy and presents new ideas/results. I would suggest moving some of this to the "Results" section or renaming this as the "Discussion" and presenting a summary of the main points in the "Conclusions."

- It would be interesting to see the difference between the 2C and 1.5C scenarios relative to the difference between the 2C and the present. That is, what fraction of the increase with the 2C scenario occurred after 1.5C?

- What role might model resolution play in the comparisons of extreme temperatures between models?

TECHNICAL COMMENTS:

- I recommend changing "apparently contradicts" in line 406 with "would seem to contradict", given the statement in lines 416-417.

---

## Referee Comment (RC2) · Anonymous Referee #2 · 18 Dec 2017

[11pt]article epsfig color

*Review of "Changes in extremely hot days under stabilized 1.5ºC and 2.0ºC global warming scenarios as simulated by the HAPPI multi-model ensemble"* (by Dr Michael Wehner and co-authors (manuscript number 10.5194-ESD-2017-89)

The paper "Changes in extremely hot days under stabilized 1.5ºC and 2.0ºC global warming scenarios as simulated by the HAPPI multi-model ensemble" uses the HAPPI

large ensemble simulations to examines changes in temperature extremes. Annual maxima of averaged three consecutive days temperature (TX3x) are modelled with a GEV distribution. TX3x 20-year return levels are compared between different warming targets (1.5$^o$C and 2$^o$C) and present climate. This study is very interesting and merits publication after the minor point below has been addressed.

Specific comment:

- I would suggest to make some hypothesis testing in order to check if TX3x 20-year return values are significantly different between different warming levels and present climate. Please, in case these differences are not significant across some region, I would suggest of highlighting the grid points with no-significant differences in each relative figure.

---

## Author Comment (AC1) · 31 Jan 2018

Responses are in red.

Anonymous Referee 1

GENERAL COMMENTS:

In this manuscript, Wehner et al. use a novel set of climate model ensembles to compare extreme temperatures under scenarios with 1.5 and 2 degrees of stabilized warming. I think this manuscript presents a useful and relevant assessment of the dif-

ferences in a metric of extreme temperatures between the two scenarios, but would benefit from additional explanations, particularly of the methods. We feel that a stationary GEV treatment has become rather standard in this class of analysis. We have added the following text to the paper: "Originally introduced by Zwiers and Kharin (1998) and Kharin and Zwiers (2000) to provide statistically rigorous projections of future extreme temperature and precipitation, such GEV analyses, both stationary and non-stationary, are now widespread throughout the literature including recent assessment reports of the International Panel on Climate Change (Seneviratne et al., 2012; Collins et al. 2013). The particulars of the details of the GEV analysis used in this study are described in the Supplementary material of Tebaldi and Wehner (2017)."

SPECIFIC COMMENTS: - In the abstract, I think it would be worth mentioning the range of increases compared to the All-Hist scenarios as well. We modified the abstract as follows: "We find that over land this measure of extreme high temperature increases from about 0.5 to $1.5^oC$ over present day values in the $1.5^oC$ stabilization scenario depending on location and model. We further find an additional 0.25 to $1.0^oC$ increase in extreme high temperatures over land in the $2.0^oC$ stabilization scenario."

- Lines 72-75: Is the prescribed SST field determined from the CMIP5 ensemble mean? To clarify, we have rewritten and expanded this paragraph as follows: "Prescribed SSTs for the $1.5^oC$ stabilization scenario are obtained by adding the average climatological change over the periods 2006-2015 to 2091-2100 from the multi-model CMIP5 RCP2.6 ensemble to the observed 2006-2015 SSTs. For the $2^oC$ stabilization scenario, a weighted sum of the RCP2.6 and RCP4.5 ensemble average changes over the same period is constructed to be exactly $0.5oC$ warmer in the global mean than the $1.5^oC$ experiment. ... While the changes to SST and sea ice concentrations defining the stabilizations scenarios are identical for each HAPPI model, the actual observations used come from a variety of well established sources chosen at the discretion of the modeling groups."

- Lines 144-146: I'm having trouble seeing the importance of this sentence. We agree and have deleted this sentence.

- Can the authors expand the explanation of the benefit of using 3-day averages? We had this statement in the original text (although we had forgotten to include the citation, which is now included): "We have previously found that while long period return values of TX3x are slightly smaller than for the daily quantity, projected changes of the 3 day averages were considerably larger (Tebaldi and Wehner 2017). For this study, where we are interested in the small differences between the 1.5$^o$C and 2.0$^o$C stabilization levels, this point becomes particularly important."

We also note that from an impacts point of view, longer duration heat waves can be more damaging than isolated hot days. In fact, the following definition of a heat wave comes from Glickman, Todd S. (June 2000). Glossary of Meteorology. Boston: American Meteorological Society. ISBN 1-878220-49-7. "To be a heat wave such a period should last at least one day, but conventionally it lasts from several days to several weeks." However, we feel it outside the scope of this paper to belabor the point.

- In line 163, it is mentioned that the 20 year return values of TX3x are estimated "using a block maxima technique", but as the TX3x values are block maxima, this description provides little information. It would probably be better to say "using the Generalized Extreme Value distribution" instead. - Agreed. We changed "using a block maxima technique" to "by fitting stationary Generalized Extreme Value distributions"

- It was not clear how the ensembles contributed to the values calculated and plotted in this paper. Were the block maxima pooled across the realizations before fitting the GEV distribution or are the ensemble mean return periods plotted? Additionally, it would be helpful if the number of values used to fit the GEV distributions were stated. We revised and added the text as follows: "By pooling the block maxima variable, TX3x, across both years and ensemble members, the extreme value time series are equivalent in length to the product of these two dimensions. As both the historical and stabilization

periods are a decade, this results in extreme value sample sizes for the HAPPI models that are 10 times longer than the number of realizations in the 3rd column of table 1, ranging from 500 (MIROC5) to 5000 (CAM4). These large sample sizes of the HAPPI models (table 1) ensure that uncertainty due to the fitting of statistical distribution is negligible. The coupled model results (CESM1) are taken directly from Sanderson et al. (2017) which used periods of three decades to compensate for the smaller ensemble size."

- Line 190: The first paragraph of the Results section can be removed, as much of this seems confusing and distracting and the relevant information is presented again at the beginning of the next paragraph. Agreed. We have deleted this paragraph and adjusted the grammar of the first sentence of the next paragraph accordingly.

- In all of the six-panel figures, the authors should make a distinction between CESM and the HAPPI models. I accept an argument for including the CESM model information, but these simulations are not directly comparable to the others. We added this sentence: "Results shown in figures 1-4 from the coupled ocean-atmosphere model, CESM1, are shown for illustrative purposes only and are not directly comparable to the HAPPI models as the experimental protocols are necessarily different."

- Line 233: Perhaps mention the later discussion of the changes in mean vs extremes. We added this sentence: "However, there are significant regional differences between models, as shown below in figure 6."

- For Figure 6, the caption description does not match what is listed in the text (at line 362), which causes confusion in the understanding of what is being plotted. Thank you. We modified this sentence slightly to (changes in blue here): "Figure 6 shows the difference between changes in 20 year return values of TX3x and changes in hot season average temperatures for the 2.0$^o$C stabilization scenario relative to the historical period."

- Line 419-420: Has this been tested? If not, perhaps rephrase to present this statement with less confidence. We deleted the sentence as it was speculative.

- The "Conclusions" section is lengthy and presents new ideas/results. I would suggest moving some of this to the "Results" section or renaming this as the "Discussion" and presenting a summary of the main points in the "Conclusions." We broke the section into two pieces with the majority of it labeled "Discussion".

- It would be interesting to see the difference between the 2C and 1.5C scenarios relative to the difference between the 2C and the present. That is, what fraction of the increase with the 2C scenario occurred after 1.5C? We have considered this request but elect not to include such a figure for two reasons. First, the actual temperature difference between the two stabilization scenarios is already shown in figure 4 and is easily interpretable. Second, comparison with the historical period is complicated by the differences in sulfate aerosol forcing between models as discussed at length in the paper. This would be reflected in such a fractional increase figure. This modified sentence clarifies this behavior: "As a result, the differences between stabilization scenarios in extreme temperature changes shown in figure 4 are less spatially heterogeneous and more similar between models than changes relative to the present day (figures 1 and 3)."

- What role might model resolution play in the comparisons of extreme temperatures between models? It is likely that model resolution could play an important role in areas of high topography and may be as important model formulation in such regions. We plan on examining this issue in a separate paper using a single model (CAM5.1) run at two different resolutions (25km and 100km). HAPPI simulations are nearly completed for that study. Until we have that study completed, it is not possible for us to make an informed comment on this matter for the models included in this paper.

TECHNICAL COMMENTS:

- I recommend changing "apparently contradicts" in line 406 with "would seem to contradict", given the statement in lines 416-417. Done.

---

## Author Comment (AC2) · 31 Jan 2018

Responses in red.

The paper "Changes in extremely hot days under stabilized 1.5oC and 2.0oC global warming scenarios as simulated by the HAPPI multi-model ensemble" uses the HAPPI large ensemble simulations to examine changes in temperature extremes. Annual maxima of averaged three consecutive days temperature (TX3x) are modelled with

a GEV distribution. TX3x 20-year return levels are compared between different warming targets (1.5oC and 2oC) and present climate. This study is very interesting and merits publication after the minor point below has been addressed.

Specific comment: I would suggest to make some hypothesis testing in order to check if TX3x 20-year return values are significantly different between different warming levels and present climate. Please, in case these differences are not significant across some region, I would suggest of highlighting the grid points with no-significant differences in each relative figure.

Rather than pick a significance level, we have elected to plot the standard error estimates directly. As the ensemble sizes are quite large, uncertainty from the statistical fitting procedure is minimal. However, uncertainty from internal variability remains and can be quantified. This subject will be dealt with a forthcoming paper that is underway. We have added the following text to the main body as well as a third figure to the appendix (reproduced below). "Standard errors obtained from the method of Hoskins and Wallis (1997) are shown to be small in figure A3 of the appendix. Generally, these error estimates are less than $0.15^oC$ with the largest values towards the higher Northern latitudes. Variability in CanAM4 is higher than the other HAPPI models but is generally less than $0.25^oC$. Standard error estimates in the CESM are of a similar magnitude but are not directly equivalent. Most of the changes in figures 1-4 are interpreted as at least at the likely level in the IPCC calibrated language (Mastrandrea et al. 2010)."

The new figure caption is:

Figure A3: Standard error estimates of 20 year return values of TX3x ($^oC$) in the $1.5^oC$ or $2.0^oC$ HAPPI simulations. Upper left: CAM4. Upper right: CanAM4. Middle left: ECHAM6. Middle right: MIROC5. Lower left: NorESM1. Lower Right. CESM.

CAM4

CanAM4

ECHAM6

MIROC5

NORESM1

CESM1

-0.45  -0.35  -0.25  -0.15  -0.05  0.05  0.15  0.25  0.35  0.45

**Fig. 1.**